# The Health Benefits of Egg Protein

**DOI:** 10.3390/nu14142904

**Published:** 2022-07-15

**Authors:** Michael J. Puglisi, Maria Luz Fernandez

**Affiliations:** Department of Nutritional Sciences, University of Connecticut, Storrs, CT 06269, USA; maria-luz.fernandez@uconn.edu

**Keywords:** egg protein, malnutrition, weight loss, athletes, sarcopenia

## Abstract

Once the general public accepts that dietary cholesterol is not a concern for cardiovascular disease risk, foods that have been labeled as high-cholesterol sources, including eggs, may be appreciated for their various other dietary components. One of the nutrients in eggs that deserves further discussion is egg protein. Egg protein has been recognized to be highly digestible and an excellent source of essential amino acids, with the highest attainable protein digestibility-corrected amino acid score. Egg protein has been shown to decrease malnutrition in underdeveloped countries, possibly increase height in children, and protect against kwashiorkor. Egg protein has been demonstrated to be important to skeletal muscle health and protective against sarcopenia. Egg protein also can decrease appetite, resulting in a reduction in the caloric intake from the next meal and weight reduction. Other protective effects of egg protein addressed in this review include protection against infection as well as hypotensive and anti-cancer effects.

## 1. Introduction

Eggs are a controversial food item due to their relatively high content of dietary cholesterol. The 2015 United States Department of Agriculture (USDA) Dietary Guidelines for Americans removed the upper limit for dietary cholesterol, emphasizing that eggs no longer need to be restricted [1]. Furthermore, many studies conducted in healthy [2,3] and non-healthy populations [4,5] clearly indicate that eggs do not increase the biomarkers associated with heart disease risk. In contrast, eggs contain several nutritional components which protect against chronic disease, including lutein, zeaxanthin, choline, vitamin D, selenium, and vitamin A [6]. An additional beneficial nutrient present in eggs is protein. It is well established that eggs are one of the best dietary sources of high-quality protein, with all of the essential amino acids, and are recognized for their high biological value [7]. In fact, eggs are considered to be the perfect protein source, serving as the standard for comparison for other protein sources [8] According to the 2018 USDA National Nutrient Database, one large egg contains 6.3 g of protein distributed between the yolk and white portions (3.6 g in egg white and 2.7 g in egg yolk) [9].

The composition and function of protein in egg whites and egg yolk varies and is associated with biological and antimicrobial activities [9]. For example, in egg whites, ovotransferrin binds metal ions, ovomucin has antiviral properties, and lysozyme disrupts the cell walls of Gram-positive bacteria [6]. Similarly, egg yolk contains lipovitellin, representing the HDL subfraction in the egg yolk and phosvitin, which protects against lipid oxidation by binding heavy metal ions [8]. One thing that needs to be taken into consideration regarding eggs is the presence of egg allergies that have been identified in children [10], which are due mostly to the proteins present in the egg whites [11].

Egg protein confers various health benefits which are advantageous for individuals across the life spectrum [11]. In the next sections, the role of egg protein in supporting skeletal muscle synthesis, protecting against sarcopenia, decreasing appetite, preventing protein deficiency, and contributing to normal child growth will be discussed.

## 2. Egg Protein and Prevention of Malnutrition in Children

As previously stated, eggs contain many nutrients and bioactive components which may help to prevent chronic disease, including protein. Protein quality is assessed by the protein digestibility-corrected amino acid score, or PDCAAS; the higher the value of the PDCAAS, the better the protein meets the requirements for all essential amino acids in regard to amounts provided and digestibility [12]. For example, for children aged from 6 months to 5 years, the PDCAAS for eggs is 118%, compared to 92–94% for meat and fish, 90–93% for soy, and 35–57% for cereals including rice, wheat, and corn [12]. Egg protein is also the lowest-cost protein, compared to many other sources, making it affordable for low socioeconomic groups [13]. However, it is important to remember that egg whites contain protease inhibitors that might decrease amino acid digestibility [14]. However, these can be easily destroyed by heat, highlighting the importance of cooked eggs being a better source of bioavailable protein.

The consumption of eggs varies between continents and countries. For example, in Latin America, a greater proportion of young children consume eggs when compared to Asian or African countries [15]. In Nepal, pregnant women have reported that religion is one of the most important reasons why they do not consume eggs [16]. In contrast, in Ethiopia, eggs are believed to be an important food source for children to grow physically and mentally [17]. In children from other underdeveloped countries, eggs and other high-protein sources are not introduced early, leading to kwashiorkor and other forms of protein malnutrition [18]. It has also been demonstrated that the introduction of eggs after weaning leads to significantly lower cases of kwashiorkor [19]. Maternal consumption of eggs during lactation may enhance the composition of nutrients in breast milk and promote the development of breastfed children [20].

Other studies have demonstrated an improvement in physical growth with egg supplementation in children from China [21] and reduced anemia and better growth outcomes in infants with mothers encouraged to consume eggs in education sessions compared to those in a control group that did not receive education in rural Sichuan, China [22]. At the end of the study, significantly more mothers from the intervention group concluded that egg yolk should be the first complementary food fed to infants [22]. Furthermore, Iannotti et al. [23] recently reported that when children in Ecuador consumed one egg per day for 6 months, the prevalence of stunting was reduced by 47% and underweight was reduced by 74% therefore assisting in achievement of normal growth. These astonishing results highlight the importance of high-quality protein in preventing malnutrition.

There have been some attempts to improve the consumption of eggs in several countries to decrease malnutrition, especially in children at risk. For example, the homestead food production model (HFP) has been utilized to improve both young infant and child nutrition when the change in behavior is associated to livestock development [24]. This program has been successful in Cambodia, where increases in egg intake were reported for children younger than 5 years of age [25]. In Indonesia, a large marketing campaign promoting the intake of eggs and leafy vegetables was very successful in increasing egg consumption with noted increases in the concentrations of vitamin A and plasma retinol [26]. In contrast, some attempts at specific programs increasing egg consumption in Bangladesh [27] and Egypt [28] have not proven to be successful. Because of the evidence that children’s physical growth can be improved [19,20,21,22,23] and malnutrition can be prevented with increased egg consumption, there is the need to promote these messages to families who live in poverty and make them aware of the low cost of this remarkably complete protein and the reported benefits for infants and children.

## 3. Egg Proteins and Skeletal Muscle Health

Skeletal muscle plays a major role in overall health, one that is often only appreciated in relationship to improvement of body composition [29,30]. Enhancing skeletal muscle health improves insulin sensitivity [31] and physical function [32] and reduces the risks of prehypertension and hypertension, cardiovascular disease [30,33], osteoporosis, and the overall risk of all-cause mortality [34]. Resistance exercise training, also important for skeletal muscle health, increases the need for protein to support elevated protein synthesis for hypertrophy and to counteract muscle protein breakdown resulting from training [35]. While the RDA of 0.8 g protein/kg body weight/day meets the needs for nitrogen balance for the majority of people, nitrogen balance is not an indicator of optimal muscle health and does not take the increased needs associated with exercise training into consideration [35]. Experts recommend a range of 1.4–2.0 g/kg body weight/day of protein for people who exercise regularly [36]. This range should be adequate for most people involved in regular physical activity. However, protein needs may be higher for athletes completing heavy training to maximize muscle hypertrophy [37]. Additionally, negative effects on kidney function have not been found after consumption of over 3 g protein/kg body weight per day or in subjects consuming ~2.5 g protein/kg body weight for 1 year [37,38].

Given that approximately 73.1% of American adults are overweight or obese [39], a focus on protein sources that are high in quality but also not excessive in calories is necessary. Along with promoting satiety (as discussed below), eggs are moderate in calories, containing 72 calories for one large egg with 6.3 g of high-quality protein [40]. Therefore, consuming eggs provides a simple, inexpensive way for people to meet their protein needs. This is of particular concern for athletes working towards higher amounts of protein, as it is critical to optimize the efficiency of the protein ingested.

The quality of protein ingested is salient because muscle protein synthesis is lower after consumption of plant proteins than animal proteins, which is a result of a lack of one or more essential amino acids, lower leucine content, and lower digestibility [41]. Eggs, meat, and dairy are digested at a rate above 90%, compared to a range of 45–80% for plant proteins [42]. In fact, eggs have been reported to be the most digestible protein source by the World Health Organization, measured as 97% [43], compared to 95% for dairy and 94% for meat, and have the highest PDCAAS score, as indicated above [12]. Soy and wheat protein are also converted to urea at higher rates, reducing the amount of protein available to stimulate muscle protein synthesis [44]. Research by Norton et al. [45] with rats confirms that eggs provide a greater anabolic stimulus. In this study, muscle protein synthesis was greater with the consumption of diets containing protein from eggs or whey as compared to soy and/or wheat and rats consuming wheat protein had 20% higher body fat than the other groups after 11 weeks [45].

Research specifically assessing the effects of eggs on muscle protein synthesis is limited, but evidence from the field pointing to the importance of protein quality and data from Moore et al. [46] assessing muscle protein synthesis with egg consumption is promising. These authors provided 0, 5, 10, 20, and 40 g of whole egg protein to healthy young men after leg resistance exercise training on five separate occasions. Researchers found that 20 g of egg protein was sufficient for maximizing muscle protein synthesis, in line with the findings from Witard et al. [47] that 20 g of whey protein also optimally stimulated muscle protein synthesis. This is also consistent with the International Society of Sports Nutrition recommendation of 20–40 g of high-quality protein per dose.

Two studies incorporating eggs with resistance training failed to find benefits for protein metabolism, but this may have been related to protein dose. Hida et al. [48] compared a daily 15 g egg white protein supplement to a carbohydrate supplement daily for 8 weeks for female athletes. No differences were reported for body composition or muscle strength. However, total protein intake for the athletes consuming the egg white protein supplement was suboptimal (1.2 g/kg body weight/day) and the 15 g dose was likely not adequate for stimulating muscle protein synthesis. Similarly, Iglay et al. [49] reported no difference in skeletal muscle mass or body composition for older adults consuming a high-protein diet focusing on eggs as a primary protein source when compared to a diet lower in protein. However, the high-protein diet provided 1.2 g protein/kg body weight per day, below the recommended intake, and timing of protein to space it out equally throughout the day was not emphasized for participants.

In contrast to the studies above, a daily snack consisting of 15 g egg white protein and 18 g sugar included with 5 weeks of resistance training increased skeletal muscle mass and strength and reduced fat mass for young men in a study by Kato et al. [50]. Contributors to these positive results likely include the subject population and the fact that absolute protein intake was 1.3 g/kg body weight/day, which, although only slightly higher than the other studies, is closer to the recommendations for people taking part in resistance training.

Recent research has assessed the effects of whole egg protein at an optimal dose of 20 g or more in the context of a diet providing an adequate amount of dietary protein (1.4 g/kg body weight or greater) with promising results. Bagheri et al. [51] provided resistance-trained males three whole eggs or an isonitrogenous source of eggs whites immediately after resistance training for 12 weeks (three times per week). The subjects maintained a protein intake of ~1.5 g/kg body weight throughout the study. The researchers reported increased strength, as determined by knee extension and handgrip strength, elevated testosterone, and lower body fat percentage for both groups, with larger increases for the group consuming whole eggs. Consumption of whole eggs and egg whites led to similar increases in body mass, anaerobic power, growth hormone, and IGF-1 after training. A separate study by the same group [52] reported similar improvements in body composition, strength, and hormonal environment with consumption of whole eggs or egg whites after resistance training with a protein consumption of ~1.4 g/kg body weight per day, further displaying the benefits of egg protein for improved skeletal muscle health.

Research with rats points to the potential benefits of egg protein over the high-quality milk protein casein. Matsuoka et al. [53] fed male rats diets containing 20% egg white protein or casein for 4 weeks, with pair feeding to match intake. The researchers reported greater average carcass protein mass and gastrocnemius leg muscle weight and lower carcass triacylglycerol and abdominal fat mass in rats fed the diet containing egg white protein. The authors attributed these changes to a few potential mechanisms, including greater net protein utilization for egg white protein (95%) as compared to casein (70%), placing an emphasis on the high digestibility of egg protein [54]. In addition, the research group previously reported reduced lipid absorption with egg white protein consumption [55], and egg white proteins have been shown to inhibit lipase activity [56] possibly contributing to the reduction in abdominal fat. Further research is necessary to assess the effects of long-term intake of egg protein compared with other high-quality proteins on skeletal muscle and body composition.

The amount of the essential amino acid leucine provided in a protein source is critical since it is the strongest stimulator of muscle protein synthesis [57]. In fact, the leucine content of a protein source alone can independently predict a food’s ability to stimulate muscle protein synthesis [45]. The recommended amount of leucine for maximal stimulation of muscle protein synthesis is between 700–3000 mg [34]. One egg provides ~500 mg of leucine in just 72 calories, making this an excellent option for meeting this proposed “leucine threshold” [34].

In order to meet meal requirements to maximize muscle protein synthesis, the incorporation of whole foods that are nutrient dense and contain high-quality protein is vital, particularly at breakfast. The yolk of an egg provides ~40% of its protein [58]. Given that nutritionists and other health professionals have often recommended for many years to consume only the egg white of an egg, a substantial amount of high-quality protein is often removed, further reducing protein intake at meals that already provide inadequate amounts of protein. Additionally, Van Vliet et al. [58] reported that when matched for protein content, consumption of whole eggs resulted in greater stimulation of myofibrillar protein synthesis than egg whites after resistance exercise for young men. Bagheri [51] also found greater strength gains, as described above, when comparing whole egg consumption to egg whites after resistance training. This may be related to the other contents of the yolk, such as phospholipids, microRNAs, or other micronutrients [59]. Additional research by Evans et al. [60] indicates increased muscle protein synthesis for older adults after supplementation with fortetropin, a complex of protein and lipids made from egg yolk that has been shown to stimulate the mammalian target of rapamycin (mTOR). The lipid portion of this supplement, along with factors found in egg yolk may be important for greater stimulation of muscle protein synthesis and subsequent strength gains. The benefits of high-quality protein provided in the yolk of an egg combined with evidence linking the intake of eggs to improved plasma lipoproteins and potentially a reduction in risk for cardiovascular disease [61] point to the need for more attention to be placed on consuming whole eggs versus just egg whites.

Many American adults are overweight or obese and are attempting to lose weight to optimize health. Researchers have determined that retention of lean body mass with weight loss requires protein intakes above the Recommended Dietary Allowance (RDA), possibly up to 2.3–3.1 g protein/kg body weight per day [34]. Pasiakos et al. [62] reported that consuming 1.6 g protein/kg body weight/day (twice the RDA) for 1 month preserved fat-free mass and enhanced fat loss during weight loss, when compared to consuming protein at the level of the RDA. Consumption at triple the RDA (2.4 g protein/kg body weight/day) had no added benefit over twice the RDA, indicating that 1.6 g protein/kg body weight/day may be appropriate for healthy, physically active individuals attempting to lose weight.

The Physical Activity Guidelines for Americans include a recommendation to complete 150 min or more of moderate intensity exercise or 75 min of vigorous exercise [63]. Although this exercise performed by most people would likely not reach the intensity of training completed by athletes, muscle damage associated with this exercise likely requires additional protein above the RDA. Combined with recommendations to include strength training twice per week, it is clear that to ensure excellent skeletal muscle health, protein intake for physically active individuals, not just competitive athletes, should be above the RDA, with the high-quality protein from whole foods, like eggs, that should be spaced evenly throughout meals at least three times per day. While the intake of high-quality protein, such as that provided by eggs, has been shown to be crucial for skeletal muscle health, many of the studies discussed in this review included whole eggs or egg whites. Therefore, it is possible that other factors present in eggs are advantageous. Given that the research discussed above [59,60] showed greater benefits with the consumption of the yolk, components such as lipids, micronutrients, and other factors in the yolk may enhance the positive effects of egg protein.

### Egg Proteins and Sarcopenia

As adults age, gradual losses of muscle mass, strength, and endurance occur, resulting in an inability to perform the activities of daily life, with a subsequent loss of independence and hampered mobility [64,65]. This process can begin as early as 30 years of age, with reductions in muscle mass of up to 8% per decade and significantly more rapid losses after 70 years of age [66]. Sarcopenia is linked to sedentary lifestyle, anabolic resistance, low-grade inflammation, and vitamin D deficiency [64,67]. The loss of function and reduction in mobility can exacerbate chronic disease, as sarcopenia increases the risk of type 2 diabetes, cardiovascular disease, some cancers, Alzheimer’s disease, and other forms of dementia [68,69].

Anabolic resistance is the term given to the lowered sensitivity of skeletal muscle to amino acids or dietary protein that occurs with aging [70], placing older individuals at a greater risk for muscle loss and sarcopenia. The factors put forth as potential causes of this resistance include low-grade inflammation and reduced physical activity, as well as insulin resistance, which lowers the anabolic effects of this hormone [71]. Two primary lifestyle changes that can be utilized to prevent, treat, or attenuate progression of anabolic resistance and sarcopenia are adding weight training to sensitize skeletal muscle to protein, especially in that 24-hour post-exercise “anabolic window” [71] and increasing dietary protein intake to overcome this resistance [72]. Most researchers in the field recommend 1.0–1.3 g of protein/kg of body weight/day for older adults to counter catabolism, with a potential need of up to 2.0 g of protein/kg of body weight/day for those with severe injury, illness or protein malnutrition [72].

Most adults in Western countries consume adequate amounts of protein overall. However, intake is often suboptimal for older adults. National Health and Nutrition Examination Survey (NHANES) data from 2003–2004 indicate that 10–25% of older adults (51 years of age and older) consume less than the RDA of 0.8 g of protein/kg body weight/day, and up to ~9% of older adults consume less than the Estimated Average Requirement (EAR) of 0.66 g/kg body weight/day [73]. Analysis of data collected between 2001 and 2014 indicates that this issue is consistently present for older Americans. Since it is generally accepted that the RDA is likely suboptimal for older adults, a large percentage of older individuals are at significant risk for sarcopenia. In the Newcastle 85+ Study, dietary protein intake was determined to average 1.0 g/kg body weight/day for over 700 individuals 85 years of age and older [74]. Older adults in residential care facilities are at particular risk for protein malnutrition and resulting sarcopenia. Tieland et al. [75] estimated that protein intake was below the EAR (half the amount of dietary protein recommended by experts) for 35% of older adults in these facilities.

The type of protein provided in a meal is critical for sarcopenia. Aligning with the positive effects mentioned above in regard to muscle protein synthesis for animal proteins, various studies have reported that animal proteins, including milk, beef, and eggs, are uniquely effective in promoting anabolism in elderly populations [76,77,78]. For example, elderly women were reported to have significantly greater net protein synthesis following a high animal protein diet compared to a high vegetable protein diet [79].

Consumption of 1.0–1.3 g/kg body weight per day or more, with 28 g of protein at each meal to maximally stimulate protein synthesis can be extremely difficult for older individuals. Diets are potentially limited by low caloric needs, poor appetite, chewing and swallowing difficulties, and pre-existing malnutrition or chronic disease, bringing issues of practicality to the forefront. Eggs are inexpensive and adding them to the diet provides a simple way to boost the intake of high-quality protein. As discussed above, eggs are a low-cost, nutrient-dense food, providing 6.3 g of high-quality protein per egg with just 72 calories and low in saturated fat relative to other sources of high-quality protein [40]. Eggs are simple to prepare, can be cooked in a short period of time, and can be prepared in many different ways, preventing boredom and reduced intake associated with lack of variety. Since eggs are soft, unlike many other high-quality protein sources, they can be prepared in ways that older individuals with poor dentition or chewing difficulties can tolerate. Eggs can be eaten at any time, but since breakfast is usually a small meal lacking in protein for older adults, adding eggs to this meal would help meet daily requirements as well as the optimal one-sitting protein amount.

Along with stimulating muscle protein synthesis, egg protein can serve to improve skeletal muscle health and prevent sarcopenia for older individuals by reducing muscle protein breakdown. In a study by Kim et al. [80], 12 subjects between the ages of 57–74 years took part in a crossover study, consuming a breakfast based on eggs and an isonitrogenous cereal breakfast. Interestingly, despite the egg breakfast containing higher quality protein, there was no difference in the stimulation of muscle protein synthesis after the consumption of the two breakfasts. However, overall nitrogen balance improved more after consumption of the egg breakfast as a result of a greater reduction in muscle protein breakdown [80].

Additionally, with the onset of the obesity epidemic in Western society, many older individuals have “sarcopenic obesity,” characterized by significant loss of muscle mass and elevation of fat mass [81]. This condition can further exacerbate anabolic resistance as a result of insulin resistance and inflammation from adiposity [82]. Weight loss without adequate dietary protein can lead to progression of muscle loss and sarcopenia. Optimally, weight loss to improve the health of older individuals with sarcopenic obesity would be done with a diet with various high-quality protein foods (as described above) and physical activity, including resistance exercise, to preserve muscle mass.

In the context of an ~1800 kcal diet, older men and women (~70 years of age) in a study by Wright et al. [83] were able to consume 1.4 g protein/kg body weight by eating three eggs at breakfast and an egg-based snack that provided 10 g protein in the afternoon. Interestingly, these subjects normally consumed ~1.0 g protein/kg body weight, and ~2000 calories per day, and were able to lose the same amount of weight as subjects consuming 0.8 g protein/kg body weight, while preserving more muscle mass. The eggs helped to boost protein intake at breakfast so that the recommended amount of protein to maximize protein synthesis was reached at all three meals for the high-protein group. On the other hand, this threshold was only met at dinner for the subjects consuming protein at the RDA. Wright et al. [83] were able to fill the need for food sources that provide a substantial amount of high-quality protein in a small to moderate amount of calories by adding eggs to the diet.

The addition of resistance exercise along with the threshold intake of 1.4 g protein/kg body weight could be crucial for prevention of loss of muscle mass in older individuals. Ullevig et al. [84] reported an improvement in strength for older women given an egg white supplement for 3 weeks as compared to a maltodextrin placebo; however, lean body mass losses were similar between groups. The supplement increased protein intake, but most participants were still below 1.2 g/kg and did not take part in resistance training to maintain lean body mass.

Finally, animal research points to the potential benefits of individual amino acids for lean body mass maintenance. Kido et al. [85] fed rats casein, albumin, or egg white protein for 14 days and found a significant increase in the growth of the soleus and extensor digitorum longus muscles after egg white consumption as compared to casein. Muscle arginine matched muscle growth, and the addition of arginine to the casein diet led to matching muscle growth from egg white protein. The authors contributed this increase in muscle growth to arginine’s role in stimulation of insulin and IGF-1. Eggs are also a good source of the amino acid cysteine, which is required for synthesis of glutathione, an important antioxidant [86]. Given that sarcopenia is characterized by increased inflammation and oxidative stress, a regular source of cysteine to maintain glutathione antioxidant activity may be helpful for reducing oxidative stress in aging. Preliminary data from an animal study by Jiayu et al. [87] found that providing geriatric mice with a diet supplemented with egg white protein increased glutathione in the heart, reduced oxidative stress and inflammation, and subsequently reduced oxidative damage to the myocardium. More research is needed, but the potential antioxidant effect of cysteine in eggs is promising. The effects of egg protein on body composition and the practical implications are shown on Table 1.

## 4. Egg Proteins, Immunity and Protection against Chronic Disease

As mentioned above, egg proteins have demonstrated anti-microbial and immuno-protective properties, which could have potential therapeutic applications [9]. Immunoglobulin Y (IgY), the equivalent of the mammal IgG, protects the developing chicken in a manner similar to IgG’s protection of the fetus in humans [88]. One of the valuable contributions of IgY is its use for passive immunization to treat and prevent human and animal diseases [89]. For example, egg yolk IgY has been shown to decrease intestinal inflammation in mice infected with *Salmonella typhimurium* [90]. Other protective components of egg proteins against infection include lysozyme, which hydrolyzes peptidoglycans in bacterial cells walls [91]; avidin, which binds biotin and prevents its utilization by bacteria [92]; phosvitin, which has been shown to have a protective effect against *Streptococcus* bacteria [93]; and ovotransferrin, whose antimicrobial activity is due to the trapping of iron that impedes bacterial growth [94].

Egg proteins have also been shown to protect against chronic disease. A hypotensive effect has been shown in rats consuming peptides from albumin [95]. It has also been demonstrated that for egg protein to be effective, the protein needs to be hydrolyzed to release the hypotensive peptides [96]. In agreement with these observations, a study conducted in a young healthy population demonstrated that one egg per day resulted in a decrease in blood pressure when compared to no eggs per day [3]. Ovotransferrin has also been recognized for its anticancer and antihypertensive properties [94]. Furthermore, lisozome has also been shown to protect against inflammatory bowel disease [97].

Controversial data exist regarding the association between egg intake and cancer in different meta-analyses. While some studies found an association between egg intake and increased incidence risk [98], others failed to find this connection [99,100]. However, lysozyme from egg white has also been shown to exert anti-cancer activity in both in vitro and in vivo conditions [101,102]. Other studies have shown egg proteins to have a protective effect against colon cancer [103,104], either via cysteine peptidase inhibitors isolated from chicken eggs [103] or showing a direct suppression of colon tumors with egg yolk proteins [104]. To summarize, the proteins in eggs not only protect against skeletal muscle loss and protein-related malnutrition, but they are also a source of biologically active components which may protect against hypertension, inflammatory bowel disease, and cancer.

## 5. Eggs Protein, Satiety, and Weight Loss

High-protein foods that have been identified to decrease appetite become important in our current society, where the annual cost associated with obesity will increase to $900 billion by the year 2030 [105]. For example, foods characterized by a high satiety index (SI) have been shown to reduce energy intake in the following meal [106]. In addition, foods that suppress ghrelin, the hormone associated with appetite stimulation [107], may be another important way to control obesity by inducing a possible decrease in hunger.

Eggs have been shown to suppress appetite and to decrease plasma ghrelin levels [108,109]. In the case of comparing two distinct breakfasts (eggs versus oatmeal in healthy subjects), significantly lower concentrations of plasma ghrelin were observed during the egg period [108]. In a randomized crossover study, 25 men aged 20–70 years, were given either an egg- or bagel-based isocaloric breakfast to determine the effects of egg protein on postprandial appetite hormones, plasma insulin, and glucose [109]. After 7 days, subjects repeated the protocol with the alternate breakfast. Blood was taken every hour for a total of 5 h in order to measure the values for these biomarkers after each breakfast food. All 25 subjects had greater decreases in appetite following the egg breakfast as measured by visual scales. In addition, using dietary records for assessment, these individuals consumed less kcal both in their next meal and in the next 24 h with egg intake [109]. The decreases in appetite were correlated with lower area under the curve for plasma glucose, insulin, and ghrelin, potentially contributing to the high SI of eggs. The lowering of plasma ghrelin levels could be related to the high protein level of eggs [110]. In agreement with this study, Vander Wal et al. [111] reported greater satiety with an egg breakfast, when compared to a non-egg breakfast.

When eggs are incorporated into meal plans, they can enhance weight loss [112]. One hundred and fifty-two men and women were given either an egg or a bagel breakfast in a weight loss intervention [112]. Greater weight loss (65%), greater reduction in waist circumference (34%), as well as a greater reduction in body fat (10%) was observed with the egg breakfast. Additionally, there were no significant changes in blood cholesterol between the groups. In contrast to these findings, a recent study reported no differences in weight in college students consuming an egg versus a non-egg breakfast. However, in spite of the additional 400 mg of dietary cholesterol per day, the values for plasma cholesterol were not different as compared to a non-egg breakfast [113].

## 6. Conclusions

It is clear that egg protein has a number of beneficial effects that protect humans across the life spectrum. Eggs are a low-cost protein source that might protect against malnutrition [18,19,20,21,22,23] in children, potentially improve skeletal muscle [46,50], and prevent sarcopenia in older adults [83]. Egg protein has also been shown to protect against infection [7,90,91], act as a hypotensive agent [3,33,34], and even protect against cancer [101,102], Finally, egg protein is associated with reductions in appetite and weight loss [108,109,112]. The main beneficial effects of egg protein are presented in Figure 1.

## Figures and Tables

**Figure 1 nutrients-14-02904-f001:**
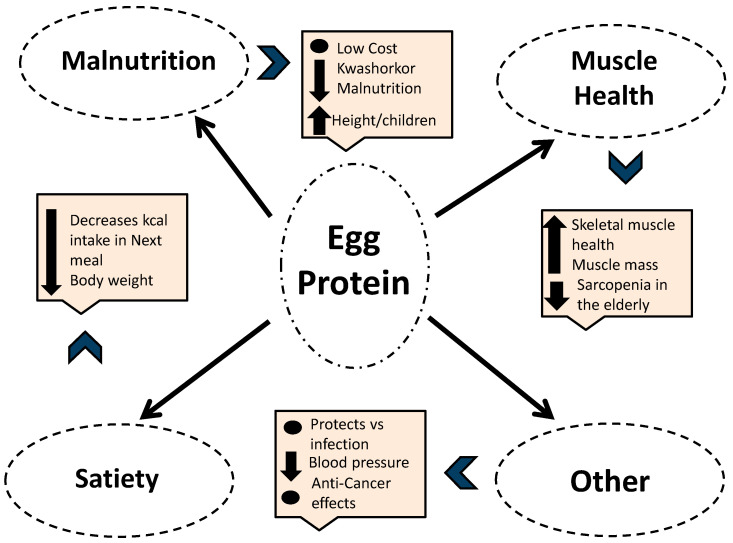
Egg protein has been shown to decrease malnutrition, improve muscle health, increase satiety, and therefore contribute to weight loss. It has other additional benefits including protection against infection, decreases in blood pressure, as well as an anti-cancer effect.

**Table 1 nutrients-14-02904-t001:** Amounts of consumed egg protein, effects on skeletal muscle health, and practical implications.

Subject Population	Weight Management or Loss	Protein Intake (g/kg Body Weight/Day or g in Single Dose)	Protein Metabolism/Body Composition	Practical Implications	Authors
Healthy young men	N/A	0, 5, 10, 20, 40 g egg protein after resistance exercise	Maximal muscle protein synthesis reached with 20 g dose	20 g egg protein is optimal single dose for young males	Moore et al. (2009) [46]
Young female athletes	Slight reduction, no difference between groups	1.0 (daily 15 g egg white protein) vs. 1.2 for 8 weeks	No differences in body composition or strength changes between groups	15 g dose egg protein as part of 1.2 g/kg/day is not sufficient for female athletes	Hida et al. (2012) [48]
Young males	No change	1.3 (daily 15 g egg white protein) with and without exercise vs. no supplement with exercise for 5 weeks	Egg white protein and resistance exercise increased skeletal muscle mass and strength, reduced fat mass	15 g dose egg protein as part of 1.3 g/kg/day diet may be sufficient for young resistance trained males	Kato et al. (2011) [50]
Resistance-trained young males	Increased for both groups	1.5 for both groups, 3 whole eggs vs. isonitrogenous source of egg whites	Improved body composition for both groups, larger reduction in body fat percentage and greater increases in strength for the whole egg group, trend of greater lean body mass gains with whole eggs	Three whole eggs or an isonitrogenous amount of egg whites is potentially beneficial for trained males consuming 1.5 g/kg/day of protein, whole eggs may have added benefits	Bagheri et al. (2021) [51]
Older men and women	No change	0.9 vs. 1.2 (focused on egg protein) for 12 weeks	No difference in body composition or skeletal muscle	1.2 g/kg/day, with a focus on eggs is not sufficient for older individuals	Iglay et al. (2009) [49]
Older men and women	Loss for both groups (−3.3%)	0.8 vs. 1.4 with 3 eggs per day for 12 weeks	Lean body mass preserved with 1.4, reduced for 0.8	Sarcopenia is countered by 1.4 g/kg/day with eggs vs. 0.8 during weight loss	Wright et al. (2018) [83]

## Data Availability

Not applicable.

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
