# Peer review of "The Health Benefits of Egg Protein"

_nutrients, 2022, doi:10.3390/nu14142904_

Round 1
Reviewer 1 Report
The present manuscript by Puglisi and Fernandez is a review on the benefits of egg protein in human health regarding malnutrition in children, skeletal muscle health and sarcopenia, immunity and protection against chronic disease, and satiety and weight loss. This review of literature is interesting and is illustrated with a number of studies including relatively recent works. The health benefits of eggs is a topic also found in another review entitled “The Golden Egg: Nutritional Value, Bioactivities, and Emerging Benefits for Human Health” (doi: 10.3390/nu11030684), recently published in Nutrients in 2019, and focused on the nutritional value of eggs, bioactivities of egg proteins (antimicrobial, immunomodulatory, antioxidant, anticancer, antihypertensive) and the factors affecting egg quality. Although similarities exist (especially regarding egg protein bioactivities), the two reviews have well-differentiated scopes. It is however surprising that this article is not cited in the manuscript of Puglisi and Fernandez.
Specific comments:
-The egg white contains very large amounts of protease inhibitors (antinutritional factors) that may impair the activity of digestive enzymes and delay the degradation of egg proteins. Some of these antinutritional factors can be denatured by heat. Therefore, the “high digestibility” of egg protein should be rather considered with regards to the consumption of cooked eggs. I think this detail is important to mention because eggs (or parts of eggs) can also be eaten raw.
-Contrasting with their numerous benefits, a brief note on the allergen risk of egg proteins could be mentioned in the introduction.
-Some bioactive properties of egg proteins seem to be unveiled after hydrolysis (e.g. antihypertensive activities, lines 333-336). This suggests that the degradation of proteins during the digestion process may release biologically-active peptides during the transit in the gastrointestinal tract. Peptides generated by proteolytic cleavage from egg proteins can potentially exhibit novel activities (antimicrobial, anticancer, antihypertensive, antioxidant, immunomodulatory…). It would be interesting to develop this aspect in the manuscript (especially in Part 4 - Egg Proteins, Immunity and Protection against chronic disease) depending on literature data available on this subject.
-A number of studies with animals or humans fed with (whole?) eggs are described in the review. The effects on health are assumed to be due to egg protein which is the main subject of the review. Is it conceivable that these effects can also be partly (or solely?) related to other egg nutrients (lipids, vitamins, minerals…) than proteins?
Minor remarks:
-Abbreviations such as RDA and EAR should be defined for non-specialist readers.
-Line 37: cobalamin is not an egg protein. There may be a confusion with ovotransferrin (previously known as conalbumin) which is an iron-binding egg protein
-Line 38: Invert “Gram positive bacteria” and “cell walls” to make sense to the sentence
-Lines 328 and 331: Gender and species names of microorganisms must be in italics. The first letter of the gender name has to be in capital letter.
-Line 328-332: The three proteins cited here (lysozyme, avidin, phosvitin) do not reflect the high diversity of antimicrobials present in the egg. Lysozyme is important in terms of activity and abundance in the egg, but the two others are less relevant when compared to other egg compounds like ovotransferrin, for instance. Examples of egg antimicrobial proteins can be found in numerous papers, including the review article mentioned above (doi: 10.3390/nu11030684).
-Line 336: I don’t think that much protein hydrolysis takes place in fried eggs. Peptides are most likely produced in the gastrointestinal tract during the digestion process by proteolytic enzymes (e.g. pepsin, trypsin…).
-Line 340-342: the statement with omega 3 might be not relevant as the review is focused on egg protein
-Line 358: I think ghrelin is rather associated with appetite stimulation, not appetite suppression
-Check correspondences between statements and cited references. Several errors seem to be present (e.g. lines 348-349, 363, 365, 370, 390…).
-Please also check the reference list: dots and spaces are present in the beginning of some references
Reviewer 2 Report
The manuscript should be improved with the new, actual literature. The references are quite old.
